# Unveiling Putative Functions of Mucus Proteins and Their Tryptic Peptides in Seven Gastropod Species Using Comparative Proteomics and Machine Learning-Based Bioinformatics Predictions

**DOI:** 10.3390/molecules26113475

**Published:** 2021-06-07

**Authors:** Viroj Tachapuripunya, Sittiruk Roytrakul, Pramote Chumnanpuen, Teerasak E-kobon

**Affiliations:** 1Department of Genetics, Faculty of Science, Kasetsart University, Bangkok 10900, Thailand; viroj.ta@ku.th; 2Omics Center for Agriculture, Bioresources, Food and Health, Kasetsart University (OmiKU), Bangkok 10900, Thailand; 3Functional Ingredients and Food Innovation Research Group, National Center for Genetic Engineering and Biotechnology (BIOTEC), National Science and Technology Development Agency, Pathum Thani 12120, Thailand; sittiruk@biotec.or.th; 4Department of Zoology, Faculty of Science, Kasetsart University, Bangkok 10900, Thailand; pramote.c@ku.th

**Keywords:** gastropod, mucus, bioactive peptides, machine-learning prediction, proteomics

## Abstract

Gastropods are among the most diverse animals. Gastropod mucus contains several glycoproteins and peptides that vary by species and habitat. Some bioactive peptides from gastropod mucus were identified only in a few species. Therefore, using biochemical, mass spectrometric, and bioinformatics approaches, this study aimed to comprehensively identify putative bioactive peptides from the mucus proteomes of seven commonly found or commercially valuable gastropods. The mucus was collected in triplicate samples, and the proteins were separated by 1D-SDS-PAGE before tryptic digestion and peptide identification by nano LC-MS/MS. The mucus peptides were subsequently compared with R scripts. A total of 2818 different peptides constituting 1634 proteins from the mucus samples were identified, and 1218 of these peptides (43%) were core peptides found in the mucus of all examined species. Clustering and correspondence analyses of 1600 variable peptides showed unique mucous peptide patterns for each species. The high-throughput k-nearest neighbor and random forest-based prediction programs were developed with more than 95% averaged accuracy and could identify 11 functional categories of putative bioactive peptides and 268 peptides (9.5%) with at least five to seven bioactive properties. Antihypertensive, drug-delivering, and antiparasitic peptides were predominant. These peptides provide an understanding of gastropod mucus, and the putative bioactive peptides are expected to be experimentally validated for further medical, pharmaceutical, and cosmetic applications.

## 1. Introduction

Snails and slugs are gastropods in the phylum Mollusca, which is the second most diverse animal phylum, and are found in many habitats, including gardens, forests, freshwater ponds and rivers, mangroves, and oceans. Gastropods have been classified into four major groups based on mitochondrial genome comparisons: Patellogastropoda (true limpets), Vetigastropoda (abalone), Caenogastropoda (previously called prosobranch and includes, e.g., periwinkles, cone snails, and slipper shells) and Heterobranchia (also known as pulmonate or opisthobranch, including, e.g., terrestrial and aquatic gastropods, such as slugs and snails) [1]. The gastropods have diverse feeding modes. Most are nocturnal herbivores and scavengers that are ecologically important to nutrient recycling, while some gastropods feed on other animals. Snails are food of many carnivorous animals, including humans, e.g., *Cyclophorus* sp., *Pomacea canaliculata* (golden apple snail), and *Helix pomatia*. Certain gastropods can adapt to new environments and quickly reproduce, enabling them to be distributed widely and become agricultural pests, e.g., *Achatina fulica* (giant African snail), *Cryptozona siamensis* (Siam Cryptozona), and *Pomacea canaliculata* (golden apple snail). Several gastropods can be intermediate hosts of parasites, i.e., rat lungworm (*Angiostrongylus cantonensis*), which can be transferred from *A. fulica* and *H. pomatia* to rats, which are their definitive hosts [2,3].

Gastropods secrete mucus from epithelial glands covering their body, as previously reviewed by Smith [4]. The mucus facilitates snail feeding, locomotion, skin protection, hydration, adhesion, and defense against infection and predators [4,5]. Some land snails, e.g., *H. aspersa*, secrete specific chemicals in mucus to communicate and to attract mates for reproduction [6]. Another study in *A. fulica* showed that the secreted mucus has healing properties on damaged shells [7]. Gastropod mucus typically contains mostly water, and the mucus viscosity is formed by complex interactions between proteins and an abundance of charged carbohydrates, including glycoproteins and glycosaminoglycans (GAGs), which are formed into complex proteoglycans, and small peptides in association with allantoin, glycolic acids, and minerals [8]. Studies on the mucus of *A. fulica* found several novel compounds, such as acharan sulfate, which is a unique glycosaminoglycan in this snail, achacin (L-amino acid oxidase enzyme, similar to escapin and aplysianin secreted from ink of the sea slugs), achatinin (9-O-acetyl sialic acid (9-O-AcSA) binding lectin), and mytimacin-AF (a cysteine-rich antimicrobial peptide) [9,10,11,12,13]. The mucus content varies according to the function and secretory structure of the glands and the gastropod species. For example, a study by Smith et al. [14] showed that adhesive mucus of the marine limpet snail *Lottia limatula* had a higher protein content than the non-adhesive mucus. This group identified a high-molecular-weight cross-linked protein (118 kDa) in the adhesive mucus. In contrast to that in limpets, in marsh periwinkle snails (*Littoraria irrorata*), the adhesive mucus had nearly an equal proportion of proteins and carbohydrates [15].

Variation in the mucous proteins was also shown in different gastropod species. Pawlicki et al. [15] found 82, 97, and 175 kDa proteins in the mucus of the land snail *H. aspersa*, while a smaller protein of 15 kDa was identified in the land slug (*Arion subfuscus*) mucus. The mucus content can vary due to different secretory gland and tissue structures. A structural study of the secretory glands in *Helix aspersa* showed four types of secretory glands: mucus gland types A and B for secreting acid mucopolysaccharides, calcium glands, and protein glands [16]. Similarly, the mucus glands, calcium glands, and channel cells were also found in the dorsal epithelium of the land slug *Ariolimax columbianus* [17]. However, some land slugs have both mucus glands and protein glands but lack calcium glands, which may result in the secretion of very watery mucus [18]. Once released as packets from the glands, these components react and cross-link to form a viscous substance. According to these contents and promising properties, gastropod mucus has been widely used as a bioactive compound in pharmaceutical and cosmetic products. The bioactive antimicrobe, anticancer, regenerative, and wound healing properties of gastropod mucus have been previously studied and reviewed [19,20,21,22]. However, despite this gastropod diversity, the mucus from only a few gastropod species, i.e., *A. fulica* and *H. aspersa,* have been studied and commercially used.

As the mucus secreted from different gastropod species might contain similar major GAG and proteoglycan types, the mucous proteins and peptides are complex and are not fully characterized. Therefore, in this study, we aimed to extract mucus proteins and identify putative bioactive peptides from the mucus of seven commonly found or commercially valuable gastropods using 1D SDS-PAGE and peptide sequencing by nano LC-MS/MS together with bioinformatics predictions to determine their putative functions. The peptide data provide a basic understanding of snail mucus protein content, and the potential bioactive peptides identified are expected to be experimentally characterized for medical, pharmaceutical, and cosmetic applications.

## 2. Results

### 2.1. Mucous Proteins Separated on The 1D-SDS Polyacrylamide Gels and Identified by Mass Spectrometry

The mucus samples had fewer Coomassie brilliant blue bands than indicated by the silver-stained gels (Figure 1 and Figure 2A). Additional protein bands were apparent after silver nitrate staining. Seven gastropod species in this study had different mucus protein patterns. The mucus protein patterns of *A. fulica*, *C. siamensis*, and *S. siamensis* were more complex than those of the other species. Analysis of the gel images by the GelAnalyzer program revealed the numbers of protein bands on the gels (Figure 2A). The program found 13, 15, and 17 countable bands from the mucus proteome of *C. siamensis*, *A. fulica*, and *S. siamensis*, respectively, and between 10 and 12 bands for the other species.

The results from the mass spectrometric analysis led to the identification of 45,597 putative protein accession numbers as determined by comparing all the peptides from the mucus proteomes of these gastropod species to those of the mollusc proteins in the NCBI database. Seventeen percent (7867 accession numbers) of these putative proteins were identified by at least two peptides, and an average of 7761 ± 20 protein accession numbers was found for individual species. All the gastropod species shared 1591 core proteins that were identified only on the basis of the core peptides, while 1775 dispensable proteins were identified according to the peptides in some of the species. However, 4501 protein accession numbers were obtained for both core and dispensable peptides. After duplicated proteins were removed, 1634 proteins were functionally annotated using Gene Ontology (GO) terms by the Blast2GO program (Figure 2B and Appendix A). Some of the proteins (415/659 proteins) were annotated under the cellular localization category as anatomical cellular entities localized at the membrane and 34/659 proteins were localized at the extracellular region, while 38/144 proteins in protein-containing complexes were in the dynein complex, cytoskeleton, and collagen trimer. The majority of the mucus proteome (1126 proteins) was assigned to cellular and metabolic processes within the biological process category. Nine-hundred and forty-two proteins were involved in the regulation of biological processes, including signaling and responses to stimulus, while 235 proteins were related to adhesion and locomotion. In terms of molecular function, 961 proteins are involved in several binding activities, including binding to protein, ATP, nucleic acid, calcium, and zinc, with fewer involved in catalytic (501 proteins) and transporter (129 proteins) activities (Appendix A).

### 2.2. Comparative Mucus Peptides of Seven Gastropod Species

The results from the mass spectrometric analysis of tryptic-digested mucus proteins led to the identification of a total of 2818 different peptides in the mucus of the seven gastropod species (Appendix A). As shown in Figure 2C, the mucus of *P. canaliculata* and *S. siamensis* had the highest number of peptides (2415 peptides). The mucus of *C. siamensis* had the fewer peptides (2360 peptides). An average of 1.36 ± 0.56% of the peptides across the seven gastropod species had relative abundance values greater than or equal to 20 (the darkest green colour in Figure 3B) compared to the average abundance values of all the peptides (13.16 ± 0.45). Results from the analysis of the physicochemical properties of these peptides revealed several short peptides with low molecular mass (Figure 3C,D) and similar hydrophobic and charge properties (Figure 3E–G). The majority of the peptides in the mucus of the seven gastropod species were slightly hydrophobic and tended to have a high number of positively charged amino acids.

Of the 2818 peptides, 1218 (43%) were core peptides found in the mucus of all the gastropod species analyzed, while 1600 variable peptides (57%) appeared in certain species (Figure 3A). Hierarchical clustering of these 1600 variable peptides revealed unique mucous peptide patterns in each gastropod species (Figure 3A). More than one-half of these variable peptides were found in five (323 ± 8 peptides on average) and six (677 ± 11 peptides on average) gastropod species. An average of 176 peptides appeared in fewer than five gastropod species (127 ± 5, 36 ± 3, 11 ± 3, and 2 ± 1 peptides in four, three, two, and one species, respectively). Further, the data from k-mode clustering was used to identify 28 different patterns of the 1600 variable peptides within these seven-gastropod species (Figure 4A). Six of these patterns (patterns 2, 4, 8, 9, 14, and 25) were based on more than 100 variable peptides and accounted for 71.04% of them. These peptide patterns were tabulated in a contingency table and shown as a balloon plot in Figure 4B. The sizes of the red circles represent the number of peptides that shared the same pattern within the same gastropod species. The results from the correspondence analysis of these patterns showed that the first two dimensions could explain 45.04% of the variation in the data. Inclusion of the latter three dimensions covered 93.84% of the variance. The biplot in Figure 4C shows global patterns of the variable peptides (blue circles) and gastropod species (triangles). The peptide patterns close together share similar profiles: for example, patterns 7, 21, and 27; patterns 8 and 25; and patterns 22 and 28. The square cosine scores in Figure 4D indicate the quality representation of the peptide patterns across different gastropod species based on the first two dimensions. The peptide patterns of *A. fulica*, *C. fulguratus*, and *S. siamensis* were distinguishable on these two dimensions, whereas the patterns of *C. siamensis*, *P. canaliculata*, and *H. distincta* were similar and could be better distinguished on the basis of the other dimensions.

All 2818 peptides were assessed by machine-learning classifiers developed in-house by the authors for predicting 20 properties of the bioactive peptides. Fourteen (antibacterial, antibiofilm, anticancer, antifungal, antihypertensive, antiparasitic, anti-inflammatory, antiviral, cell-communicating, cell-penetrating, drug-delivering, quorum-sensing, toxic, and tumor-homing peptides) and eleven (antibacterial, antihypertensive, antiparasitic, anti-inflammatory, antiviral, cell-communicating, cell-penetrating, drug-delivering, quorum-sensing, toxic, and tumor-homing peptides) bioactive properties were predicted by the k-nearest neighbor (kNN) and random forest (RF) methods, and eleven shared properties were obtained by both methods. The performance of the kNN predictors (95.5% accuracy, 96.4% sensitivity, and 95.3% specificity) were slightly better than the RF predictors (95.2% accuracy, 95.1% sensitivity, and 95.2% specificity) (Appendix A). The kNN and RF predictors yielded the results with high confident interval, averaged no information rate of 70%, and had acceptable *p*-values, indicating better performance of the models over the no information rate. Figure 5A presents an overview of the putative bioactive properties among these 2818 peptides, and the associated values are presented in Figure 5C. Light blue shading indicates the bioactive properties predicted by one method, while dark blue shading indicates those predicted by both the kNN and RF methods. The consensus peptides of the predicted bioactive peptides contained 1089 peptides, and 969 of these peptides had putative antihypertensive properties (Figure 5C). The consensus peptides also contained 75, 47, 28, and 14 peptides with drug-delivering, antiparasitic, antiviral, and cell-communicating properties, respectively (shaded in dark blue in Figure 5A,C). These 1089 consensus peptides consisted of 56 peptides with dual bioactive properties and two peptides with triple bioactive properties. When considering all possible predicted putative bioactive peptides, some of these peptides have as many as seven bioactive properties. The peptide numbers are indicated in Figure 5D above the square bars.

The consensus predicted bioactive peptides appeared across the mucous peptides of seven gastropod species, differing slightly in terms of peptide number (Figure 5B). Antihypertensive (839 ± 10 peptides on average), drug-delivering (65 ± 4 peptides on average), and antiparasitic (40 ± 2 peptides on average) peptides were the top three putative properties enriched in the mucous peptides of all the species. The fourth and fifth top consensus properties were antiviral (21 ± 3 peptides on average) and cell-communicating (13 ± 1 peptides on average) peptides. The remaining bioactive properties were associated with in only a few species, e.g., one antibacterial, 2 ± 1 anti-inflammatory, 6 ± 1 cell-penetrating, one toxic, and 4 ± 1 tumor-homing peptides.

## 3. Discussion

The mucus proteins of seven commonly found or commercially valuable gastropod species in this study varied in terms of molecular mass and tryptic peptides. The SDS-PAGE separation of the land slug *Lehmannia valentiana* mucus yielded complex protein profiles from at least 18 protein bands [23], similar to the number of protein bands obtained in the present study which found 17 protein bands from the mucus of the land slug *S. siamensis*. The highest number of the protein bands could partly contribute to the highly viscous and elastic properties of this slug mucus to protect the shell-less body. The band profiles were nearly consistent across the replicates suggesting species-conserved patterns of the mucus proteins. However, the mucus protein profiles could also be altered in different physiological stages of the gastropods, as previously shown by Smith and Morin [24]; Li and Graham [25]; or by post-translational modification, i.e., glycosylation. The marsh periwinkle snail *Littorina irrorata* produces adhesive mucus with high concentrations of 36 kDa and 41 kDa proteins compared to the trail mucus, which had fewer proteins [24]. The epiphragm mucus, which is secreted by the vineyard snail *Cernuella virgata* during hibernation, contains a higher concentration of protein (particularly the 86 kDa epiphragmin protein) [25]. This epiphragm mucus proteomes from three snail species (*C. virgata*, *Cochlicella barbara*, and *Theba pisana*) also differed. These results suggested that genetic and environmental background influences the production of mucous proteins, and possible regulatory processes might control and alter the mucus content in response to environmental and internal stimuli, as supported by the finding in this study of several proteins with regulation and signaling functions in the mucus.

After the protein separation step, the data from mass spectrometric analysis was used to sequence all possible tryptic peptides in the mucus from these seven gastropod species, which had not been previously elucidated. Proteomics approaches are popular for use in the identification of all possible proteins in gastropods, e.g., novel proteins in the proteome of *Arion vulgaris* [26] and the egg perivitelline fluid of *P. canaliculata* [27], alteration of reproductive proteins in the reproductive organ proteome of *Lymnaea stagnalis* after different endocrine treatments [28], and the shell proteomes of the nacre and prismatic layers of the marine gastropod *Haliotis laevigata* [29]. Very few and limited proteomics studies have been used to thoroughly characterize the mucus proteome of gastropods, and this study elucidated the complexity and similarity of the mucus proteomes, which comprised 1634 proteins in total. Espinosa et al. [30] used similar gel-based proteomics to identify 205 proteins that were differentially expressed in the mucus collected from mantle, gills, and labial palps of the bivalve *Crassostrea virginica*, suggesting appropriate use of the gel-based proteomics as the gold standard method to study the mucus proteome. The currently available genomes of certain molluscs enabled the identification and comprehensive description of the mucus proteomes of these seven gastropod species. Analysis of the mucus proteomes provided molecular evidence supporting gastropod mucus functions. These extracellular and membrane proteins could be important for environmental and signal sensing, which trigger appropriate responses in terms of the amount, content, and rheology of the mucus secreted. However, the lack of genomes for some of these seven gastropods resulted in the discovery of many proteins that were unknown or hypothetical, suggesting that results from further omics analyses may help explain their functions.

Data from the peptide analysis revealed similarities in the mucus obtained from these seven gastropod species and indicated that nearly 50% of the core peptides may be shared through their evolutionary relationship and essential to mucus function; that is, they are core proteins of mucins and of other proteoglycans [31]. The variable peptides were clustered into 28 different patterns (Figure 5A) and could be used to distinguish the mucus peptides in some gastropod species, i.e., *A. fulica*, *C. fulguratus*, and *S. siamensis* (Figure 5C). The reason for the pattern differences may be explained by genetic distinctiveness or regulation of protein expression and secretion, as well as by epithelial cell types. These components were synthesized and secreted from different types of mucous glands that line the foot tissues. Greistorfer et al. [32] identified five mucous glands in *H. pomatia* that differed from those in *H. aspersa*, in which only four types were found. These glands produce acidic glycosaminoglycans and acidic proteins; therefore, having different types of mucous glands may affect the proportion of these peptide variants in the mucus. These glands could secrete different peptides/proteins by exocytotic processes, as explained in the proteomics study of the airway mucus glands in humans [33]. Our results suggest that the peptide patterns within the mucus could possibly be developed as markers for detecting mucus sources and quality.

Several putative bioactive peptides were predicted from the data on the gastropod mucus peptides based on the machine-learning-derived classifications, including antihypertensive, drug-delivering, antiparasitic, antiviral, cell-communicating, antibacterial, anti-inflammatory, cell-penetrating, toxic, and tumor-homing peptides. Several antimicrobial and anticancer peptides were previously isolated from molluscs, for example, the antibacterial peptides mytimacin-AF from *A. fulica* [13] and Bb-AMP4 from *Bellamya bengalensis* [34], the antifungal peptide MytM from *Mytilus galloprovincialis* [35], and the anticancer tripeptide from *Tegillarca granosa* [36]. This study found that certain bioactive properties have not been reported and are expected to be further examined. Putative antihypertensive, drug-delivering, and antiparasitic bioactive peptides were highly abundant within the mucus peptides. These positively charged short peptides could be endogenous peptides that were tryptic cleaved and released from precursor gastropod proteins [37,38]. The known antihypertensive peptides from various food sources help regulate blood pressure; for example, valine- and isoleucine-containing peptides can inhibit angiotensin converting enzyme (ACE) and reduce high blood pressure or osmotic pressure of the interstitial fluid [39,40]. The presence of proline, tryptophan, and phenylalanine at the C-terminus or in branched aliphatic amino acids at the N-terminus of peptides also enhances ACE inhibition [41,42]. These peptides may play a role in maintaining osmotic balance between epithelial layer and the mucus covering it to prevent tissue dehydration. Several antihypertensive peptides were also found in the lysates of fish skin, including in skin mucus [42]. Many of these peptides were also cleaved from the extracellular and membrane proteins (e.g., cadherin, collagens, integrin, mucin, and proteoglycan protein), supporting their protective roles in mucus. However, the antihypertensive function of the mucus peptides remains unproven in gastropods, and their potential role in osmotic balance will need to be further tested. The drug-delivering peptides shared similar properties with cell-penetrating, tumor-homing, and cell- communicating peptides; they are short, positively charged, and hydrophobic, with well-defined secondary structures (α-helix or β-sheet structure) [43]. At low concentrations, these peptides penetrate the cell membrane in vivo and in vitro without using chiral receptors, to cause membrane damage [44] or to deliver bioactive molecules (proteins, peptides, oligonucleotides, or nanoparticles) to cells of various types and cellular compartments, making them useful for medical and pharmaceutical applications [45,46]. The uptake of these peptides could be achieved through (1) direct penetration of the plasma membrane as explained by the barrel-stave and carpet models, (2) endocytosis, and (3) inverted micelle-mediation [47]. In the presence of non-tryptic-digested mucus peptides, these putative drug-delivering peptides may facilitate environmental signal sensing through the mucus to enable gastropods to respond appropriately to the surrounding stimuli. Some of these peptides were also derived from signaling and regulatory proteins (e.g., cell surface glycoprotein, exportin, laminin, spectrin, and twitchin), and their transmembrane domains could support the membrane penetrating ability of these peptides. This finding has caused questions on mucus signaling to be further examined in gastropods. The antiparasitic peptides were reported in several animals but not in the gastropod mucus. For example, the magainin peptide from the skin of the African clawed frog *Xenopus laevis* causes parasitic membrane perturbation and disrupts osmotic balance [48]. The temporin A and B peptides from the skin of *Rana temporaria* demonstrate antiparasitic activity, without haemolytic activity, against human erythrocytes [49]. None of the annotated proteins had known antiparasitic properties. These peptides can help prevent infections from nematodes in the surrounding environments.

The integration of the kNN and RF-based predictors could suggest bioactive peptide candidates from the gastropod mucus. Developing one complex predictor for all bioactive peptide properties remains challenging and is considerably difficult, particularly for the closely related peptide properties and limited experimental information. This study chose to design separate predictors for individual properties and integrated both ML methods to suggest the most probable bioactive peptide candidates. The simplicity of kNN by using only the k value and distance function would broadly scan for the peptides with similar features and easily sort them into the positive or negative results at the expense of computational cost for the large sample size. The ensemble ability of the RF method by random feature selection, which involved building multiple decision trees from the chosen features and aggregating the trees, protected robust predicted bioactive peptides from the large dataset and reduced risk of overfitting at the expense of possible bias when dealing with categorical variables. The consensus voting finally compromised the limitations. However, this predictor design could be improved by incorporating more complex algorithms and further reduction of the feature complexity. This study has opened new questions and opportunities for novel application of the gastropod mucus and expanding the exploration to other mollusc species. The peptide data will also enable the reconstruction of the mucus proteome from the gastropod species with an unavailable genome and suggest another tool representing the evolution of gastropod mucus research.

## 4. Materials and Methods

### 4.1. Sample Collection and Preparation

Seven commonly found or commercially valuable gastropod species were used in this study: two caenogastropods and five heterobranchs (Table 1). The two caenogastropods were aquatic snails (*P. canaliculata*) and terrestrial snails (*C. fulguratus*) that have an operculum to close their shell to protect themself. The five heterobranchs were separated into four terrestrial snail groups (*A. fulica*, *C. siamensis*, *H. distincta*, and *H. pomatia*) and one terrestrial slug (*S. siamensis*). All samples were collected from local ponds or public or university gardens. At least three individuals of each species were collected and maintained under conditions resembling their natural environment. All samples were not feeding the day before mucus was extracted to avoid proteins contaminated by sample faeces. All samples were bathed and cleaned with sterile distilled water before mucus extraction.

Mucus was collected from at least three biological replicates of gastropod samples. First, we placed the samples in a sterile Petri dish before dropping 2 mL of distilled water on the pedal surface. The round-end of a glass stirring rod was used to gently rub the gastropod foot until mucus was secreted for 5–10 min and dissolved in water. The times for the mucus collection slightly varied because of the gastropod size and behavior. To obtain samples from the small gastropods (noted with the asterisk in Table 1), which secrete a low volume of the mucus, the gastropods were placed on sterile Petri dishes containing 0.5–1 mL of distilled water and allowed to crawl around the plate for 10–15 min along with gentle rubbing with the round-end of the glass stirring rod. Mucus solutions were centrifuged at 10000 rpm for 3 min before separating supernatant and filtering. The mucus solution was collected in 1.5 mL microcentrifuge tubes and kept at −20°C until use.

### 4.2. D-SDS-PAGE Experiment

The total protein concentration of the mucus solution was quantified and adjusted by modified Lowry assay [50]. The assay was performed in triplicate, and the protein standard curve was generated according to bovine serum albumin (BSA) (Thermo Fisher, Waltham, MA, USA) in solution at serial concentrations between 0 and 1000 µg/µL. Each adjusted mucus solution was mixed with an equal volume of 2X tricine sample buffer (1 M Tris-HCL pH 6.8, glycerol (VWR International, Radnor, Pennsylvania, USA), sodium dodecyl sulfate (SDS) (Bio-Rad, Hercules, CA, USA), and 1 M β-mercaptoethanol (Applichem, Darmstadt, Germany)) and heated at 95°C for 5 min. The samples were centrifuged at 11,000 rpm for 5 min, and the supernatants were separated on 1D SDS-PAGE using a Tris/tricine buffer system. The tricine polyacrylamide gels were prepared manually using an Amersham apparatus (GE Healthcare, Chicago, IL, USA). The 12% resolving gels contained 0.5 M Tris-HCl/SDS buffer (pH 8.45), 10% glycerol, acrylamide/bis solution 37.5:1 (Bio-Rad, Hercules, CA, USA), 0.02% ammonium persulfate (APS) (Bio-Rad, Hercules, CA, USA), and 0.5 mM tetramethylethylenediamine (TEMED) (Applichem, Darmstadt, Germany). The 4% stacking gels contained 0.04 M Tris-HCl/SDS buffer (pH 8.45), acrylamide/bis solution 37.5:1, 0.02% APS, and 0.4 mM TEMED. Twenty microliters of each sample was loaded, and the gels were run using the Bio-Rad power supply (Bio-Rad, Hercules, California, USA). The gels were stained in Coomassie brilliant blue G (Applichem, Darmstadt, Germany) on an orbital shaker (Eppendorf, Hamburg, Germany) overnight at 75 rpm and de-stained twice with a solution containing 45% methanol (TEDIA, Carson, CA, USA) and 10% glacial acetic acid (EMD Millipore, Burlington, MA, USA) for 70 min before washing in distilled water for 15 min. The gels were visualized by Bio-Rad Gel Doc XR+ with Image LabTM software (Bio-Rad, Hercules, CA, USA) and preserved in distilled water at 4 °C.

Coomassie blue–stained gels were then counterstained with silver nitrate. Briefly, the gels were fixed in fixative solution (40% ethanol and 10% acetic acid) for 60 min and soaked in distilled water for at least 30 min or for at most overnight. The gels were washed in 0.02% sodium thiosulfate (VWR International, Radnor, PA, USA) followed by three rinses with distilled water. Then, 0.1% silver nitrate solution (VWR International, Radnor, PA, USA) was added and incubated for 20 min before the gels were washed in distilled water three times. The gels were developed by adding 3% sodium carbonate (VWR International, Radnor, PA, USA) under continuous shaking until bands appeared. Distilled water was added for 20 sec to remove excess sodium carbonate, and the reactions were stopped by adding 5% acetic acid and incubating them for 5 min. The silver-stained gels were kept in 1% acetic acid at 4 °C and visualized by the previously described software. Mucus gel images were computationally analyzed using the GelAnalyzer program version 2010 [51]. The areas of the sample lanes were defined as equal. The baseline mode was used to set the graph base, and the molecular weights for the molecular marker lane were calculated using MW calibration mode. These parameters were used to calculate the molecular weight of the bands from the sample in each lane. The estimated molecular weight and band intensity data were recorded and compared.

### 4.3. LC-MS/MS

After the protein bands were excised from the gels, the gel plugs were dehydrated with 100% acetonitrile (ACN) (Sigma-Aldrich, Burlington, MA, USA), reduced with 10 mM dithiothreitol (DTT) (Applichem, Darmstast, Germany) in 10 mM ammonium bicarbonate (Sigma-Aldrich, Burlington, MA, USA) at room temperature for 1 h and alkylated at room temperature for 1 h in the dark in the presence of 100 mM iodoacetamide (IAA) (Sigma-Aldrich, Burlington, MA, USA) in 10 mM ammonium bicarbonate. The gel pieces were dehydrated twice with 100% ACN for 5 min before the in-gel digestion. Briefly, 10 µL of trypsin solution (10 ng/µL in 50% ACN/10 mM ammonium bicarbonate) (Sigma-Aldrich, Burlington, MA, USA) was added to the gels and incubated at room temperature for 20 min before adding 20 µL of 30% ACN to cover the gels and incubating them at 37 °C for a few hours or overnight. The peptides were extracted by adding 30 µL of 50% ACN in 0.1% formic acid (FA) (Sigma-Aldrich, Burlington, MA, USA) into the gels and incubating at room temperature for 10 min on a shaker. The peptides were collected in the new tube and dried by vacuum centrifuge (Eppendorf, Hamburg, Germany) before maintained at −80 °C for mass spectrometric analysis.

The digested peptide solutions were analyzed with a HCTultra PTM Discovery System (Bruker Daltonics Ltd., Billerica, MA, USA) coupled to an UltiMate 3000 nano LC system (Thermo Fisher Scientific, Waltham, MA, USA) as previously described in E-kobon et al. [21]. Briefly, peptides were separated on a nanocolumn (PepSwift monolithic column 100 µm i.d. × 50 mm). Eluent A was 0.1% formic acid, and eluent B was 80% acetonitrile in water containing 0.1% formic acid. Peptide separation was achieved with a linear gradient of 10% to 70% eluent B for 13 min at a flow rate of 300 nL/min, which includes a regeneration step with 90% eluent B and an equilibration step with 10% eluent B. One run took 20 min. Peptide fragment mass spectra were acquired in data-dependent AutoMS(2) mode with a scan range of 300–1500 m/z, 3 averages, and as many as 5 precursor ions from the MS scan from 50 to 3000 m/z.

DeCyder MS Differential Analysis software (GE Healthcare, Chicago, IL, USA) was used for peptide quantification [52,53]. The LC-MS raw data were converted, and the PepDetect module was used for automated peptide detection, charge state assignments, and quantitation based on the peptide ion signal intensities in the MS mode. The MS/MS data were submitted in a search against the NCBI database using Mascot software (Matrix Science, London, UK) for possible protein identification [54]. The search parameters included taxonomy (molluscs, TaxId: 6447); enzyme (trypsin); variable modifications (carbamidomethyl and oxidation of methionine residues); mass values (monoisotopic); protein mass (unrestricted); peptide mass tolerance (1 Da); fragment mass tolerance (± 0.4 Da); peptide charge state (1+, 2+, and 3+); and maximum missed cleavages (1). Proteins identified by at least two peptides were submitted for functional annotation using the Blast2GO program version 5.2 [55] for functional classification into biological process, molecular function, and cellular component categories. Duplicated proteins with the same accession number or multiple orthologs in different organisms were removed before comparing the mucus proteins across the gastropod species using the in-house written R scripts.

The maximum value was used to determine the presence or absence of each identified peptide. Data normalization and quantification of the changes in peptide abundance between the samples were determined and visualized using Multiple experiment Viewer (MeV) software version 4.6.1 [56]. Briefly, peptide intensities from the LC-MS analyses were transformed and normalized using a mean central tendency procedure. Statistical tests of variance (ANOVA) for these data sets were performed, and significance was set to *p* < 0.05. Peptide sequences from each sample were added to the gastropod mucus peptide data set.

### 4.4. Proteomic Data Evaluation

The mucus peptide data from seven gastropod species were compared by the R scripts to identify core peptides that were found in the mucus of all species and variable peptides that appeared in certain gastropod species. The following physicochemical properties of the peptides were generated using our R scripts: molecular mass, number of amino acids, hydrophobicity score, di-amino acid compositions, and number of positively charged and negatively charged amino acids, which were later used as features for developing the machine-learning predictors. The data were hierarchically clustered by the hclust() function, and the clustering patterns were plotted by the heatmap.2() function. Categorical data clustering and correspondence analysis of the variable peptides were performed using k-mode clustering (kmodes() function) in the klaR package to reduce the complexity of the data [57], and the CA() function of the FactoMineR and factoextra libraries [58,59] was used to define the mucous peptide patterns and their interrelationships. The correspondence analysis results were summarized and illustrated by the fviz_ca_biplot() function of the factoextra library.

### 4.5. Bioinformatics and Machine Learning

Twenty-two in-house machine-learning classifiers (unpublished program) were developed based on k-nearest neighbor (k-NN) and random forest (RF) algorithms using R language to predict 20 functional categories of bioactive peptides, i.e., antibacterial, antibiofilm, anticancer, antifungal, antihypertensive, antioxidant, antiparasitic, anti-inflammatory, antiprotease, antiviral, cell-communicating, cell-penetrating, chemotactic, drug-delivering, haemolytic, insecticide, quorum-sensing, spermicide, surface-immobilized, toxic, tumor-homing, and wound-healing peptides from the gastropod mucus peptides. Our high-throughput prediction program was trained and tested with known bioactive peptides from 15 bioactive peptide databases that collectively contained 37,807 peptides with known bioactive activities: AOP-Pred (antioxidant [60]), APD3 (antimicrobial [61]), BaAMPs (antibiofilm [62]), BACTIBASE (antibacterial [63]), CPPpred (cell-penetrating [64]), DADP (antimicrobial [65]), dPABBs (antibiofilm and quorum-sensing [66]), HemoPI (haemolytic [67]), HIPdb (antiviral [68]), iAMP-2L (multi-functional antimicrobial [69]), ProInflam (anti-inflammatory [70]), QSPpred (quorum-sensing [71]), SATPdb (multi-functional [72]), ToxinPred (toxic [73]), and TumorHPD (tumor-homing [74]). Briefly, the known peptides were grouped by properties and labelled as positive and negative data, for example, antibacteria and non-antibacteria, and anticancer and non-anticancer. Four-hundred and seven features were used in developing kNN and RF models for each bioactive peptide property. The positive and negative data of each peptide properties were balanced by oversampling method using ovun.sample() function of the ROSE package and divided into training and test data at the proportion of 70%:30% using createDataPartition() and train_ABover() functions of the caret package. The created models were trained on the training dataset using the knn3() and randomForest() functions. The kNN models were created by using the knn3() function with k = 2 for the positive and negative data, and the RF models were created by using randomForest() function of the randonForest package with ntree = 1000. The prediction model performance was examined by using the confusionMatrix() function to record accuracy (a summation of the true positives and true negatives/the total number of predicted peptides), sensitivity (the number of correctly-predicted true positives/the total number of true positive peptides), specificity (a proportion of the actual negatives that were correctly identified), 95% confident interval (95% likelihood that the true positive lied within the range), no information rate (the accuracy achievable by frequent predicting the major class), and *p*-values (indicating whether the models offer significantly better performance over the no-information rate), and the predictors were applied to make predictions about the mucus peptides by using the predict() function. Predicted results from the kNN and RF predictors were combined to generate the consensus results. The putative bioactive peptides were compared across the gastropod species using R scripts.

## 5. Conclusions

In summary, mass spectrometric, machine learning, and bioinformatics analyses assisted in the discovery of 11 putative bioactive mucus peptides of seven gastropod species and provided a better understanding of the mucus functions in gastropods. The peptide data will also enable the reconstruction of the mucus proteome from the gastropod species with an unavailable genome and suggest another tool representing the evolution of gastropod mucus research. The integrated kNN and RF-based predictors could narrow opportunities for novel discovery of the gastropod bioactive peptides and can be expanded to other organismal species. These potential bioactive peptides are expected to be experimentally characterized and useful for pharmaceutical and cosmetic applications.

## Figures and Tables

**Figure 1 molecules-26-03475-f001:**
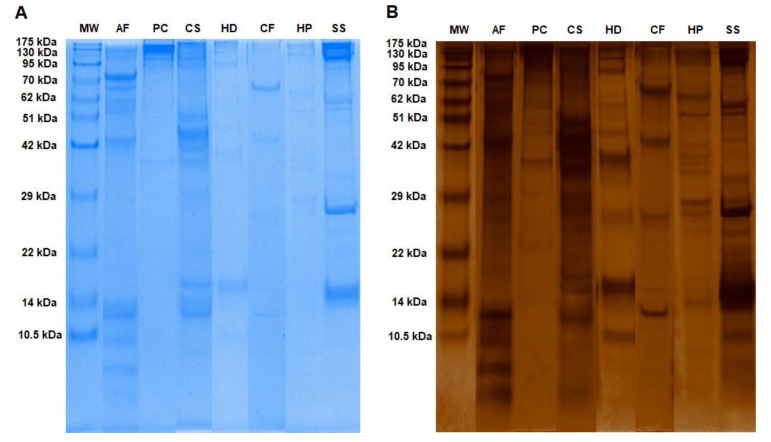
Mucus samples from seven gastropod species separated on 1D-SDS polyacrylamide gels. The gels were stained with Coomassie brilliant blue (**A**) and silver nitrate (**B**). By lane: 1, MW = protein molecular marker; 2, AF = *Achatina fulica*; 3, PC = *Pomacea canaliculata*; 4, CS = *Cryptozona siamensis*; 5, SS = *Semperula siamensis*; 6, HD = *Hemiplecta distincta*; 7, CF = *Cyclophorus fulguratus*; and 8, and HP = *Helix pomatia*.

**Figure 2 molecules-26-03475-f002:**
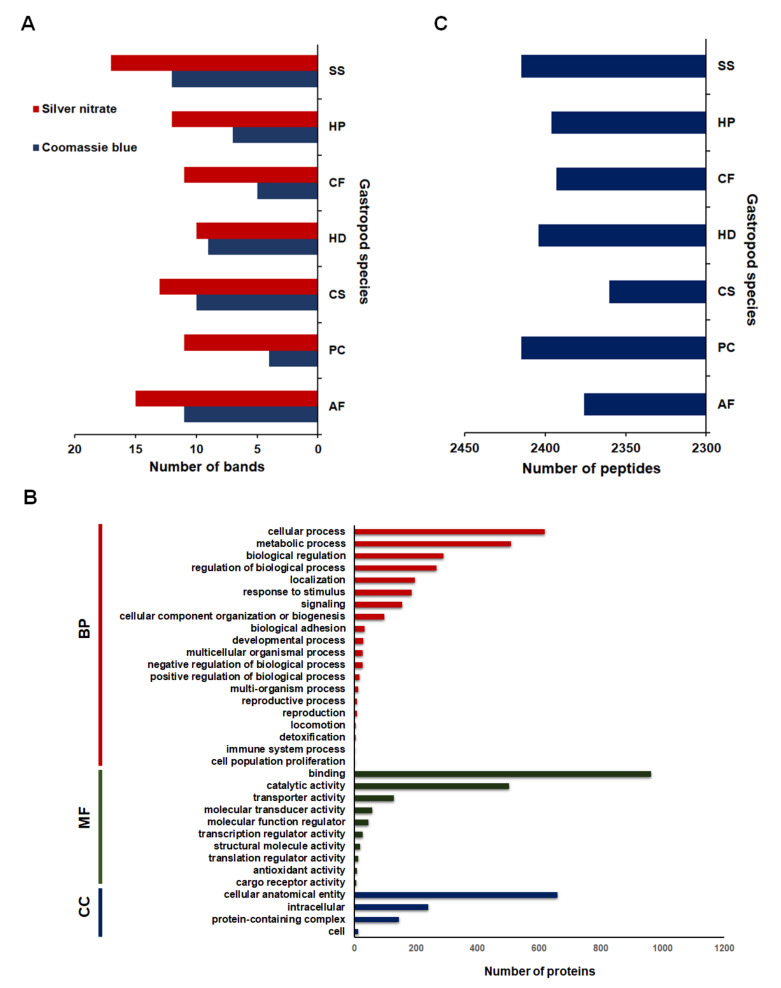
Comparative proteomes and peptides of mucus from seven gastropod species. (**A**) Number of protein bands of the mucus proteins from seven gastropod species separated by 1D-SDS-PAGE and stained by Coomassie brilliant blue (blue) and silver nitrate (red) as counted by the GelAnalyzer program. The number of protein bands is shown on the *x*-axis. The gastropod species are shown on the *y*-axis. (**B**) Functional annotation terms of the mucus proteome in three Gene Ontology categories (*y*-axis) biological process (BP), molecular function (MF), and cellular component (CC). The *x*-axis represents the number of proteins. (**C**) The number of different peptides identified in the mucus of seven gastropod species. These peptide data were obtained from the mass spectrometric analysis. The number of peptides is shown on the *x*-axis. AF = *Achatina fulica*; PC = *Pomacea canaliculata*; CS = *Cryptozona siamensis*; SS = *Semperula siamensis*; HD = *Hemiplecta distincta*; CF = *Cyclophorus fulguratus*; and HP = *Helix pomatia*.

**Figure 3 molecules-26-03475-f003:**
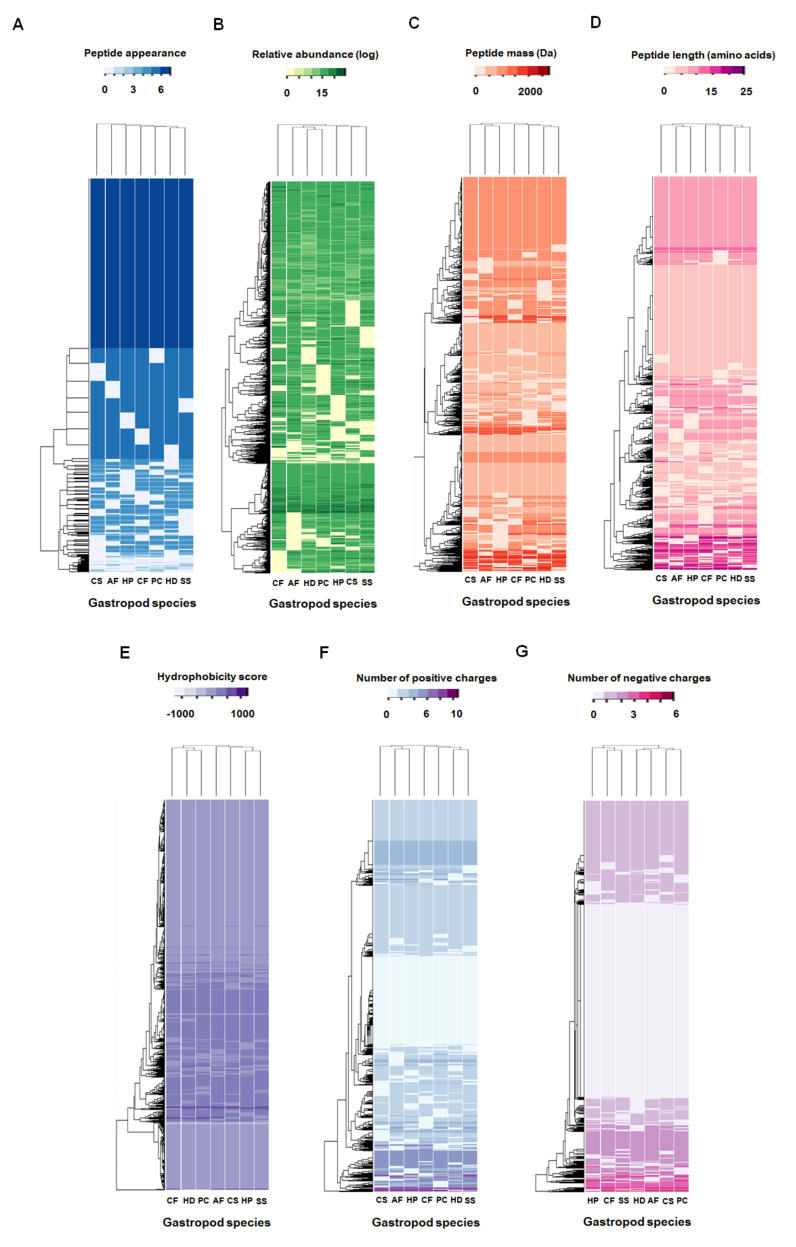
Clustering of peptides identified in the mucus of seven gastropod species. The peptides were clustered according to their presence (**A**), relative abundance (**B**), molecular mass (**C**), number of amino acids or length (**D**), hydrophobicity score (**E**), numbers of positively charged amino acids (**F**), and negatively charged amino acids (**G**). Columns represent gastropod species, and rows represent peptides. AF = *Achatina fulica*; PC = *Pomacea canaliculata*; CS = *Cryptozona siamensis*; SS = *Semperula siamensis*; HD = *Hemiplecta distincta*; CF = *Cyclophorus fulguratus*; and HP = *Helix pomatia*.

**Figure 4 molecules-26-03475-f004:**
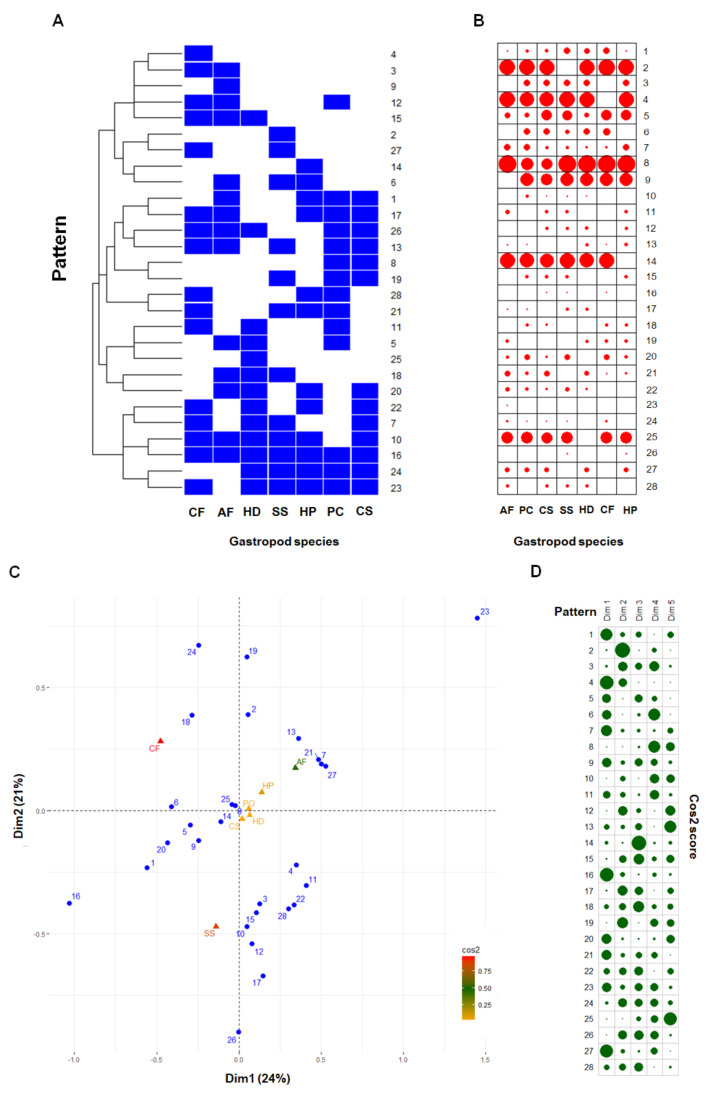
Correspondence analysis of 28 patterns of variable peptides in the mucus of seven gastropod species. Blue squares in the heatmap (**A**) indicate the presence of the peptide, and white squares indicate the absence of a peptide. The sizes of the red circles in the balloon plot (**B**) indicate the number of peptides within each pattern. The biplot (**C**) shows global patterns of the variable peptides (blue circles) and the gastropod species (triangles) according to the first two dimensions (Dim1 and Dim2). Colors of the square cosine scores indicate the gastropod species. The scores for the first five dimensions (Dim1 to Dim5) are shown in (**D**), and the sizes of the green circles represent the score level. AF = *Achatina fulica*; PC = *Pomacea canaliculata*; CS = *Cryptozona siamensis*; SS = *Semperula siamensis*; HD = *Hemiplecta distincta*; CF = *Cyclophorus fulguratus*; and HP = *Helix pomatia*.

**Figure 5 molecules-26-03475-f005:**
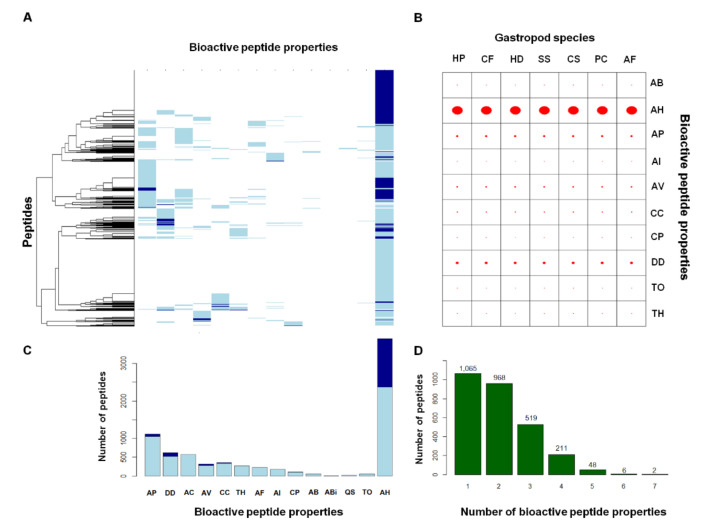
Prediction of 20 bioactive properties from the mucous peptides of seven gastropod species. The predictions were based on two machine-learning algorithms (k-nearest neighbor and random forest). (**A**) The prediction results show peptides that were predicted by either method in light blue, whereas those predicted by both methods are colored in dark blue. (**B**) Distribution of the bioactive peptides across seven gastropod species. Sizes of the red circles represent the number of peptides. (**C**) Number of the bioactive peptides classified by the properties are presented in the same order as the properties in (A). (**D**) Number of bioactive peptides (indicated by numbers on top of the bars) that had multiple bioactive properties (from one to seven). AP: antiparasitic, DD: drug-delivering, AC: anticancer, AV: antiviral, CC: cell-communicating, TH: tumor-homing, AF: antifungal, AI: anti-inflammatory, CP: cell-penetrating, AB: antibacterial, Abi: antibiofilm, QS: quorum-sensing, and AH: antihypertensive properties. The abbreviations of the gastropod species are the same as those presented in Figure 4.

**Table 1 molecules-26-03475-t001:** List of gastropod samples used in this study.

Sample	Length (cm)	Shell Diameter (cm)	Weight (g)	Location (Latitude, Longitude)
*Achatina fulica* (Heterobranchia)	7–8	7–8 **	120–140	A garden at Faculty of Science, Kasetsart University, Bangkok, Thailand (13.845416, 100572014)
*Cryptozona siamensis* (Heterobranchia)	5–6	3–4	30–35	Sakaerat Environmental Research Station Sakaerat Biosphere Reserves, Nakhon Ratchasima, Thailand (14.510218, 101.930995)
*Semperula siamensis* (Heterobranchia)	5–6	-	15–20	A tree shop, Patumthani, Thailand (14.918298, 100.9163500)
*Hemiplecta distincta* (Heterobranchia) *	7–8	5–6	120–140	Chet Khot-Pong Kon Sao Nature Study Centre, Saraburi, Thailand (14.493871, 101.161605)
*Pomacea canaliculata* (Caenogastropoda) *	6–7	4–5	80–100	A canal near Faculty of Science, Kasetsart University, Bangkok, Thailand (13.845505, 100.572566)
*Cyclophorus fulguratus* (Caenogastropoda)	6–7	4–5	80–100	Sakaerat Environmental Research Station Sakaerat Biosphere Reserves, Nakhon Ratchasima, Thailand (14.510218, 101.930995)
*Helix pomatia* (Heterobranchia)	6–7	4–5	70–90	A garden in Lisbon, Portugal (38.755233, −9.174415)

* Gastropods that were small and/or produced little mucus. ** Shell height.

## Data Availability

Not applicable.

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
