# Peer review of "Unveiling Putative Functions of Mucus Proteins and Their Tryptic Peptides in Seven Gastropod Species Using Comparative Proteomics and Machine Learning-Based Bioinformatics Predictions"

_molecules, 2021, doi:10.3390/molecules26113475_

Round 1
Reviewer 1 Report
The manuscript presents combination of 1D-SDS-PAGE, LC-MS/MS, and bioinformatic approaches for identification of bioactive peptides obtained from gastropod mucus. The content is relevant within the current proteomic field. The manuscript is appropriate for publication in Molecules with moderate revisions according to the comments listed below.
Why were the kNN and RF-based approaches selected? The benefits and disadvantages of such approaches should be mentioned.
What are the benefits of the selected algorithms in comparison to PCA?
How affected N-glycans the 1D-SDS-PAGE results?
Page 12, Table 1: The explanation of the asterisks should be added in the footnote of the table.
Chapter 4. Materials and Methods is confusing, and it is difficult to orientate in it. I recommend to divide it in subchapters according to the relevant experiments – e.g.:
4.1. Sample collection and preparation,
4.2. 1D-SDS-PAGE experiment,
4.3. LC-MS/MS,
4.4. Proteomic data evaluation,
4.5 Bioinformatics and machine learning
Chapter 5. Conclusion: The chapter is too short and very general. The benefits of the work are not summarized and presented.
Author Response
Response to reviewer 1
The manuscript presents combination of 1D-SDS-PAGE, LC-MS/MS, and bioinformatic approaches for identification of bioactive peptides obtained from gastropod mucus. The content is relevant within the current proteomic field. The manuscript is appropriate for publication in Molecules with moderate revisions according to the comments listed below.
Why were the kNN and RF-based approaches selected? The benefits and disadvantages of such approaches should be mentioned.
Ans: During prior development of the predictors, several ML algorithms were examined. The kNN and RF methods were ones with high performance, so they were selected and integrated into the predictor design in this study. The authors agreed with the reviewer comment, so the last paragraph of the discussion was added to discuss on the benefits and limitations of our selected methods. “The integration of the kNN and RF-based predictors could suggest bioactive peptide candidates from the gastropod mucus. Developing one complex predictor for all bioactive peptide properties remains challenging and is considerably difficult, particularly for the closely related peptide properties and limited experimental information. This study chose to design separate predictors for individual properties and integrated both ML methods to suggest the most probable bioactive peptide candidates. The simplicity of kNN by using only the k value and distance function would broadly scan for the peptides with similar features and easily distinguish them into the positive or negative results at the expense of computational cost for the large sample size. The ensemble ability of the RF method by random feature selection, building multiple decision trees from the chosen features, and aggregating the trees, provided robust predicted bioactive peptides from the large dataset and reduced risk of overfitting at the expense of possible bias when dealing with categorical variables. The consensus voting finally compromised the limitations. However, this predictor design could be improved by incorporating more complex algorithms and further reduction of the feature complexity. …”
What are the benefits of the selected algorithms in comparison to PCA?
Ans: Earlier the authors had used CCA (as they were mostly counting data) and PCA to categorize the mucus peptides and to identify important features for separating different bioactive peptide properties. The separation was not satisfactorily clear and certain bioactive properties were difficult to distinguish as explained previously. Therefore, the authors changed to use the targeted predictor design which produced separate predictors for individual bioactive properties by using the kNN and RF methods before integrating the results by consensus voting. The suggestion to compare our design with the PCA-based methods is interesting and the authors will possibly use PCA to reduce the feature dimension before the model building which perhaps reduce computational time and increase the performance. The authors added these benefits to the last discussion paragraph.
How affected N-glycans the 1D-SDS-PAGE results?
Ans: The glycosylation might result in the molecular mass shift of the protein bands and could alter protein profiles observed in this study. The Coomassie blue-stained gels in this study were not able to highlight the glycosylated proteins. After counter-staining the SDS-polyacrylamide gels with silver nitrate, more bands, presumably those with glycosylation, appeared. The authors might not be able to identify specific glycosylated sites of the proteins, but we are currently investigating the glycosylation by using proteomics techniques. The authors agreed with the point raised by the reviewer, so the phrase was added to the sentence of the first discussion paragraph (p. 9), “However, the mucus protein profiles could also be altered in different physiological stages of the gastropods, as previously shown by Smith and Morin [24] and Li and Graham [25], or by post-translational modification i.e., glycosylation.”.
Page 12, Table 1: The explanation of the asterisks should be added in the footnote of the table.
Ans: The footnote of Table 1 was added, “* Gastropods that were small and/or produced little mucus.”.
Chapter 4. Materials and Methods is confusing, and it is difficult to orientate in it. I recommend to divide it in subchapters according to the relevant experiments – e.g.:
4.1. Sample collection and preparation,
4.2. 1D-SDS-PAGE experiment,
4.3. LC-MS/MS,
4.4. Proteomic data evaluation,
4.5 Bioinformatics and machine learning
Ans: These suggested subheadings were added as suggested by the reviewer.
Chapter 5. Conclusion: The chapter is too short and very general. The benefits of the work are not summarized and presented.
Ans: The authors agreed with the reviewer comment, so the conclusion paragraph (p. 15) was modified to, “In summary, mass spectrometric, machine learning, and bioinformatics analyses assisted in the discovery of 11 putative bioactive mucus peptides of seven gastropod species and provided a better understanding of the mucus functions in gastropods. The peptide data will also enable the reconstruction of the mucus proteome from the gastropod species with an unavailable genome and suggest another tool representing the evolution of gastropod mucus research. The integrated kNN and RF-based predictors could narrow opportunities for novel discovery of the gastropod bioactive peptides and can be expanded to other organismal species. These potential bioactive peptides are expected to experimentally characterized and useful for pharmaceutical and cosmetic applications.”.
Reviewer 2 Report
The authors conducted a study to characterize the mucous proteins and peptides of seven gastropod species. A series of wet-lab experiments (including gel electrophoresis), followed up by AI-based classification computation work, were used to ultimately arrive at various characterizations of mucous proteins along abundance, mass, hydrophobicity, and other metrics.
Overall, the paper is well written, and the methodology is sound. There are enough details in the methods section to permit somebody to reproduce the results.
There is not necessarily anything novel being presented methodology-wise, because standard off-the-shelf tools and existing databases and software are used. But, the scale of the study is noteworthy.
Minor comments:
In the abstract, consider mentioning the accuracy of the high-throughput prediction program.
In Results, explain the importance of noting 13, 15, and 17 versus 12-12 bands for different species. What are the implications of one species having more bands than another?
In Results, you mention that duplicate proteins were removed, but you don't given enough details about that in the methods section.
In Results, you mention that "Approximately 950 proteins..." Approximately? Why not give the exact number? More importantly, these predictions and characterizations are derived ultimately from the multiple databases that were utilized. How good were the found proteins to match with the proteins in the multiple databases. They surely weren't perfect alignments/matches (you mention Blast2GO, but don't provide details on the results).
You mention 95% accuracy, sensitivity, and specificity. What were the outliers? Standard deviations? And you group both kNN and RF as both achieving those high success rates. Wasn't one necessarily a bit better or worse than the other?
Author Response
Response to reviewer 2
The authors conducted a study to characterize the mucous proteins and peptides of seven gastropod species. A series of wet-lab experiments (including gel electrophoresis), followed up by AI-based classification computation work, were used to ultimately arrive at various characterizations of mucous proteins along abundance, mass, hydrophobicity, and other metrics.
Overall, the paper is well written, and the methodology is sound. There are enough details in the methods section to permit somebody to reproduce the results.
There is not necessarily anything novel being presented methodology-wise, because standard off-the-shelf tools and existing databases and software are used. But, the scale of the study is noteworthy.
Minor comments:
In the abstract, consider mentioning the accuracy of the high-throughput prediction program.
Ans: The authors followed the reviewer comment on this point, so a sentence in the abstract was modified to, “The high-throughput k-nearest neighbor and random forest-based prediction programs were developed with more than 95% averaged accuracy and could identify 11 functional categories of putative bioactive peptides and 268 peptides (9.5%) with at least five to seven bioactive properties.”.
In Results, explain the importance of noting 13, 15, and 17 versus 10-12 bands for different species. What are the implications of one species having more bands than another?
Ans: The image analysis of the SDS polyacrylamide gels allowed the authors to compare band sizes and numbers across different gastropod species. These numeric patterns seem highly specific to the gastropod species as explained in the first discussion paragraph. However, the mucus having more protein bands than the others might imply to physicochemical properties of the gastropod mucus. For example, the mucus of the land slug, S. siamensis, had the highest number of the protein bands which could be partly inferred to the high viscosity and elastic properties of its mucus. However, this assumption must be further examined. So, the authors added additional sentence to the first discussion paragraph (p. 9), “The highest number of the protein bands could partly contribute to the highly viscous and elastic properties of this slug mucus to protect the shell-less body.”.
In Results, you mention that duplicate proteins were removed, but you don't given enough details about that in the methods section.
Ans: The authors agreed with the reviewer comment, so additional sentences were added in the method section 4.3 (p. 14) to explain the removal of duplicated proteins, “Proteins identified by at least two peptides were submitted for functional annotation using the Blast2GO program version 5.2 [55] for functional classification into biological process, molecular function, and cellular component categories. Duplicated proteins with the same accession number or multiple orthologs in different organisms were removed before comparing the mucus proteins across the gastropod species using the in-house written R scripts.”.
In Results, you mention that "Approximately 950 proteins..." Approximately? Why not give the exact number? More importantly, these predictions and characterizations are derived ultimately from the multiple databases that were utilized. How good were the found proteins to match with the proteins in the multiple databases. They surely weren't perfect alignments/matches (you mention Blast2GO, but don't provide details on the results).
Ans: The sentence was edited to “Nine-hundred and forty-two proteins were involved in the regulation of biological processes, including signalling and responses to stimulus, …. …. … (Appendix A).”. These proteins were identified from the peptide sequences obtained from the LC-MS/MS analysis. The peptide sequence data (MS/MS spectra) were searched against the NCBI protein database as described in the method section 4.3 (p. 14) using the MASCOT program. The confident protein identification was justified based on the protein score and at least two peptides matched to the protein. The authors agreed with the reviewer point that some proteins might not be well identified with perfect matches and high sequence coverage. These proteins will have to be further confirmed by the MS/MS analysis, particularly the ones with interesting functions.
After the removal of proteins with duplicated names and accession numbers by our R scripts, 1,634 proteins were functionally annotated by the Blast2GO program into three GO categories: biological process, molecular function, and cellular component. The Blast2GO results were explained in the second paragraph of the result section (p. 3). Additional Appendix Table (Appendix A) was added to the sentence to supplement Figure 2B. Sentences (in blue) after the modified one explained the protein function classification derived from the Blast2GO program.
“The results from the mass spectrometric analysis led to the identification of 45,597 putative protein accession numbers as determined by comparing all the peptides from the mucus proteomes of these gastropod species to those of the mollusc proteins in the NCBI database. … … … After duplicated proteins were removed, 1,634 proteins were functionally annotated using Gene Ontology (GO) terms by the Blast2GO program (Figure 2B and Appendix A). Some of the proteins (415/659 proteins) were an-notated under the cellular localization category as anatomical cellular entities localized at the membrane and 34/659 proteins were localized at the extracellular region, while 38/144 proteins in protein-containing complexes were in the dynein complex, cytoskeleton, and collagen trimer. The majority of the mucus proteome (1,126 proteins) was assigned to cellular and metabolic processes within the biological process category. Nine-hundred and forty-two proteins were involved in the regulation of biological processes, including signalling and responses to stimulus, while 235 proteins were related to adhesion and locomotion. In terms of molecular function, 961 proteins are involved in several binding activities, including binding to protein, ATP, nucleic acid, calcium, and zinc, with fewer involved in catalytic (501 proteins) and transporter (129 proteins) activities.”
You mention 95% accuracy, sensitivity, and specificity. What were the outliers? Standard deviations? And you group both kNN and RF as both achieving those high success rates. Wasn't one necessarily a bit better or worse than the other?
Ans: The authors agreed with the points raised by the reviewer, so 95% confident interval, no information rate, and p-values indicating whether the models offer significantly better performance over the no-information rate were added to the method section 4.5 and the Appendix Table 3, “The prediction model performance was examined by using the confusionMatrix() function to record accuracy (summation of the true positives and true negatives/the total number of predicted peptides), sensitivity (the number of correctly-predicted true positives/the total number of true positive peptides), specificity (a proportion of actual negatives that were correctly identified), 95% confident interval (95% likelihood that the true positive lied within the range), no information rate (the accuracy achievable by frequent predicting the major class), and p-values (indicating whether the models offer significantly better performance over the no-information rate), and the predictors were applied to make prediction on the mucus peptides by using the predict() function.”.
The kNN predictors had slightly better performance that those of the RFs as mentioned by the reviewer, so the authors modified a sentence in the result section (p. 8) to “The performance of the kNN predictors (95.5% accuracy, 96.4% sensitivity, and 95.3% specificity) were slightly better than the RF predictors (95.2% accuracy, 95.1% sensitivity, and 95.2% specificity) (Appendix C). The kNN and RF predictors yielded the results with high confident interval, averaged no information rate of 70%, and acceptable p-values which indicated better performance of the models over the no information rate.”.